# Unhealthy Food at Your Fingertips: Cross-Sectional Analysis of the Nutritional Quality of Restaurants and Takeaway Outlets on an Online Food Delivery Platform in New Zealand

**DOI:** 10.3390/nu14214567

**Published:** 2022-10-30

**Authors:** Nisha Mahawar, Si Si Jia, Andriana Korai, Celina Wang, Margaret Allman-Farinelli, Virginia Chan, Rebecca Raeside, Philayrath Phongsavan, Julie Redfern, Alice A. Gibson, Stephanie R. Partridge, Rajshri Roy

**Affiliations:** 1Discipline of Nutrition and Dietetics, School of Medical Sciences, Faculty of Medical and Health Sciences, The University of Auckland, Auckland 1011, New Zealand; 2Engagement and Co-Design Research Hub, School of Health Sciences, Faculty of Medicine and Health, The University of Sydney, Sydney, NSW 2145, Australia; 3Nutrition and Dietetics Group, Susan Wakil School of Nursing and Midwifery, Faculty of Medicine and Health, The University of Sydney, Sydney, NSW 2006, Australia; 4Charles Perkins Centre, The University of Sydney, Sydney, NSW 2006, Australia; 5Prevention Research Collaboration, Sydney School of Public Health, Faculty of Medicine and Health, The University of Sydney, Sydney, NSW 2006, Australia; 6The George Institute for Global Health, The University of New South Wales, Camperdown, NSW 2006, Australia; 7Menzies Centre for Health Policy and Economics, Sydney School of Public Health, Faculty of Medicine and Health, The University of Sydney, Sydney, NSW 2006, Australia

**Keywords:** food environment, online food delivery, nutrition, diet, takeaway foods, fast food, independent outlet, franchise outlet, value bundle, young adult

## Abstract

Online food delivery (OFD) platforms have become increasingly popular due to advanced technology, which is changing the way consumers purchase food prepared outside of the home. There is limited research investigating the healthiness of the digital food environment and its influence on consumer choice and dietary behaviours. This study is the first to examine the nutritional quality and marketing attributes of menu items from popular independent and franchise restaurants and takeaway outlets on New Zealand’s market leading OFD platform (UberEATS^®^). A total of 374 popular independent and franchise restaurants and takeaway outlets were identified to form a database of complete menus and marketing attributes. All 25,877 menu items were classified into 38 food and beverage categories based on the Australian Dietary Guidelines. Of complete menus, 73.3% (18,955/25,877) were discretionary. Thirty-six percent (9419/25,877) were discretionary cereal-based mixed meals, the largest of the 38 categories. Discretionary menu items were more likely to be categorized as most popular (OR: 2.0, 95% CI 1.7–2.2), accompanied by a photo (OR: 1.7, 95% CI 1.6–1.8), and offered as a value bundle (OR: 4.6, 95% CI 3.2–6.8). Two of the three discretionary mixed meal categories were significantly less expensive than their healthier counterparts (*p* < 0.001). The overwhelming availability and promotion of discretionary choices offered by restaurants and takeaway outlets on OFD platforms have implications for public health policy. Further research to explore direct associations between nutritional quality and consumers’ dietary choices is required.

## 1. Introduction

Obesity has become a global health concern and is associated with the onset of multiple chronic noncommunicable diseases (NCDs) such as type 2 diabetes mellitus, cardiovascular diseases, stroke, and cancers [1,2]. The New Zealand (NZ) Health Survey 2019/2020 conducted by the Ministry of Health found that approximately 31% of the adult population (15 years and over) were obese [3]. The escalating global prevalence of obesity is predominantly driven by dietary changes and lifestyle behaviours [1]. These behaviours include overconsumption of sweets, sugar-sweetened beverages (SSB), processed meats, high-fat dairy products, and packaged, ultra-processed snacks; and low consumption of fruits, vegetables, and wholegrains [1,4,5,6]. The modern food environment acts as a contributor to poor health of adults in NZ. One segment of the food environment is the foodservice sector, which includes dining restaurants, cafés, and fast-food or quick-service restaurants that can provide foods and drinks for immediate consumption [5].

Globally, working individuals have much busier lifestyles, hence the desire to spend less time and effort preparing food at home and increased demand for convenience [7,8]. In NZ, according to a 2018 survey, approximately 80% of children and young adults stated that they had fast-food or takeaway at least once per week [9]. According to the food price index for NZ in 2020, approximately 27% of the food budget was spent on ready-to-eat and restaurant meals which was a slight increase from 26% in 2017 [10,11]. The average Kiwi household food budget on takeaways and restaurant meals has also steadily risen from 22% in 2000 to 27% in 2020 [10]. Additionally, Auckland residents had shown to be spending the highest proportion (32%) of their food budget on takeaways as compared to the national average of 27% [10]. However, frequent consumption of takeaways and fast-foods have been associated with a poor diet quality with increased energy, total fat, sodium, and added sugars content, increasing the risk of obesity and its related comorbidities [6,12,13,14,15].

Smartphones and the internet have provided quick access to food outlets away from home, to order food directly to the consumer’s address at their convenience through online food delivery (OFD) platforms [16]. OFDs enable consumers to choose from a variety of food options from a broad range of foodservice outlets offering various cuisines [5,17,18]. Young adults (15–34 years) are reported to be the largest users of OFD platforms, constituting approximately 48% of users globally and over 25% of users in Australia and NZ [19,20]. More specifically in NZ, 47% of the reported users of OFD platforms in 2021 were young adults (18–34 years) [21].

Internationally, Uber Eats is the most popular food delivery service and remains the largest and the most popular OFD platform in Australia and NZ [20,22]. The majority of most popular food outlets on Uber Eats NZ have been categorized as discretionary and 86% of the popular menu items offered by these outlets are discretionary foods and beverages with low nutritional value [19,23]. An analysis of complete menus from popular independent restaurants and takeaway outlets in Sydney, Australia, reported 81% to be discretionary items and discretionary menu items were also more likely to be categorized as most popular [22]. Additionally, the typical nature of ordering food through OFD platforms for convenience also promotes a sedentary lifestyle, further contributing to the global burden of obesity and NCDs [1,24,25]. As a result of the COVID-19 pandemic, people across the world were forced to stay at home and have relied more on OFD services to access fast-foods and takeaways [26]. Additionally, since the pandemic, most food outlets have increased their geographical delivery distance to reach more consumers, hence increasing accessibility to food choices beyond local food outlets [27].

Multiple studies have demonstrated that OFD platforms use marketing strategies, including but not limited to appealing food images, discounts, combos, and meal deals to heavily influence consumers’ food purchases [16,22,28]. Like the effect of placing products at eye-level on supermarket shelves, “Popular Near You” and “Most Popular” options make food outlets and menu items easily visible on OFD platforms and influence consumers’ food preferences [29,30]. Currently, very limited research has been done on the marketing techniques used by OFD platforms to target consumers, highlighting a research gap.

The use of marketing strategies to promote food outlets and menu items with poor nutritional quality within the currently unregulated space of OFD platforms, highlights the need for policies and interventions. Mandatory kilojoule labelling has been implemented in some Australian states for franchise restaurants/takeaway outlets (i.e., a restaurant group that prepare and sell meals ready for immediate consumption, offered in specialized packaging, e.g., Burger King^®^) for consumers to make informed choices at point-of-purchase [5,16]. However, independent food outlets (e.g., local pizza or kebab shops), are not subject to this regulation and remain highly unregulated [31]. In NZ, in addition to OFD platforms being unregulated, neither the franchise food outlets nor the independent food outlets are subject to any mandatory labelling.

The primary aim of this study was to evaluate the nutritional quality of all menu items from popular restaurants and takeaway outlets available on Uber Eats in NZ. The secondary aim was to investigate the associations between the nutritional quality and marketing attributes of all menus.

## 2. Materials and Methods

### 2.1. Identification of Popular Franchise & Independent Restaurants and Takeaway Outlets

A previous cross-sectional study conducted in Sydney, Australia, and Auckland, NZ searched a total of 186 Auckland, NZ suburbs between 9 and 22 February 2020 to form a database of popular food outlets including independent and franchise food outlets. The identification process is described in detail in a previous study [19]. Auckland was chosen as the location of interest in this study since it is the largest city in NZ with high concentrations of young consumers (15–34 years), who are the primary users of OFD services in general [19,32]. The 10 most popular food outlets were extracted from the “popular near you” section on the Uber Eats platform, for suburbs with above-average populations of young people (>30%, 15–34 years), who are the leading users of OFD platforms [19]. The current analysis commenced in 2022, by which time 20 of the food outlets identified from this previous study ceased partnership with Uber Eats and thus were excluded. Unlike the previous study conducted in Australia [22], this NZ study also included both franchise and independent restaurants and takeaway outlets. In this study, franchise restaurants/takeaway outlets have been defined as chain stores with outlets in two or more locations with the name, brand, logo, and menu consistent across all locations (e.g., McDonalds^®^, Subway^®^, KFC^®^) [33,34]. An independent restaurant/takeaway outlet does not have any other chains and is run by an independent owner, e.g., local kebab or pizza shop [33].

Researchers were not logged into any personal Uber Eats accounts during the search to avoid personalized results and to only access publicly available data. This NZ study focused on evaluating the food outlets identified in the previous study using the Food Environment Score Tool [19,35,36,37]. Franchise stores in different locations were considered individual outlets or “unique”. Unique food outlets were classified as having a different geographical location (e.g., Subway Botany, Subway Mission Bay, etc.).

### 2.2. Data Extraction

Publicly available complete menus were extracted from the Uber Eats website on 10 September 2020 (via web scraping, ScrapingSolutions) [22]. Complete menus include all menu items available from independent and franchise restaurants and takeaway outlets as displayed on their Uber Eats webpage. Data extracted from these menus for each food outlet included the menu items’ names, descriptions, Uber Eats categories, prices, photos, nutritional information (e.g., the macronutrient profile), and any dietary labelling (e.g., vegan, vegetarian, gluten-free, etc.) if available. In the Australian study, most popular menu items were extracted for analysis from the ‘Most Popular’ section displayed at the top of each food outlet’s webpage [19,22]. However, most of the food outlets in NZ displayed “Picked-for-you” instead, hence was assumed to be the same as “Most Popular” in this study. To improve readability, the term ‘popularized menu items’ will be used in this article combining both ‘most popular’ and ‘picked for you’ categories. Table 1 provides a summary of definitions and the derivations of the data extracted.

### 2.3. Outcome Measures

This study’s primary outcome was to evaluate the nutritional quality of complete menus from popular independent and franchise restaurants and takeaway outlets available on the market leading OFD platform (i.e., Uber Eats) in NZ. The secondary outcome was to investigate the associations between nutritional quality and marketing attributes for complete menus, including popularity cue, use of photos and promotional offers.

#### 2.3.1. Nutritional Quality

All menu items from the most popular independent and franchise restaurants and takeaway outlets on Uber Eats were classified into 38 food and beverage categories based on the menu item description and/or photo provided of the menu item on the food outlet’s Uber Eats webpage, using an updated version of a classification system previously proposed for a sub-study of the MYMeals project [38]. The Australian Dietary Guidelines were used to define all menu items into Five Food Group (FFG) and discretionary classifications [19,23]. Menu items classified as FFG contain food(s) or a combination of foods from the five food groups: vegetables and legumes/beans, fruit, grain (cereal) foods, mostly wholegrain, and/or high cereal fibre varieties; lean meats and poultry, fish, eggs, tofu, nuts and seeds, and legumes/beans; and milk, yoghurt, cheese and/or alternatives, and mostly reduced fat [19,23]. Australian Dietary Guidelines has defined discretionary foods as foods that are high in saturated fats, sugars, sodium and/or alcohol and lower in dietary fibre [23]. The Australian Dietary Guidelines were used in this study for consistency between similar studies conducted previously [19,22] and because they align with the NZ Eating and Activity Guidelines [39].

The Australian Bureau of Statistics’ (ABS) principles and list for identifying discretionary foods were used to assist categorization [40]. For menu items with insufficient information such as stir-fries, without adequate detail in the description (although stir-fries have excessive sodium and would be considered discretionary if the saturated fat content is >5 g/100 g), a conservative approach was taken. These menu items were classified as a FFG type dish. However, when a menu item contained a discretionary ingredient (e.g., battered, or crumbed meats and seafood, processed meats such as sausages or bacon, katsu (fried) chicken, cream-based curries, or hot chips and SSB as part of a meal deal), then the menu item was classified as discretionary [19,22].

Menu items as part of a value bundle (i.e., meal deals and family deals), were categorized for each food item included. For example, burgers, chips, and drink as part of a meal deal or a combination menu item were all separately analysed for nutritional quality. Combination menu items in this study have been defined as menu items consisting of more than one food/beverage item. For example, menu item ‘fish and chips’ are categorized as a combination menu item as both ‘fish’ and ‘chips’ are individual food items assigned to different food categories (i.e., Meat or alternative-based mixed meal (discretionary) and Fried potato (or similar) respectively). Unlike this current NZ study, the Australian study included combination menu items and coded these items as meal deals when there was an option to purchase the individual components from the food outlet and the combination meal was available at a discounted price [22]. Combination menu items in the Australian study were only assigned a single category for nutritional quality based on the main component of the menu item. For example, ‘fish and chips’ in the Australian study were categorized as meat or alternative-based mixed meal (discretionary) only; unlike the NZ study which also categorized ‘chips’. The combination menu items contain more than one food/beverage item, hence in the NZ study, they were analysed separately to avoid an overlap of food and beverage categories in the analysis.

All SSB (i.e., beverages containing added sugars or nutritive sweeteners such as soft drinks, bubble teas, and milk or alternative-based beverages with discretionary items such as syrups and sweeteners) were classified as discretionary. For food outlets that allowed consumers to design their own meal, e.g., Subway; the nutritional analysis was based on the image and description provided on the OFD platform as a suggestion for the consumers. Furthermore, menu items that lacked significant data to enable classification into one of the groups were classified as “undetermined” as there were multiple categories they could be assigned to (e.g., “drink” or “meat dish”, with no image or description provided for classification). Additionally, some inedible menu items (e.g., cutlery) were classified as “non-consumable”. All menu items categorized as undetermined and non-consumable were excluded from data analysis. The nutritional analysis of all menu items was conducted by a student dietitian (NM). A random 20% of the data was cross-checked by a registered dietitian (RR) and there were no disagreements.

#### 2.3.2. Marketing Attributes

Marketing attributes included: popularity cue (Uber eats category of “Most Popular” and “Picked-for-you”), price, photo of menu item, value bundles, special promotions, nutritional information, and dietary labelling. The web scraping company extracted marketing attributes, excluding value bundles, special promotions, and picked-for-you categories, during data extraction from the Uber Eats website. The Uber eats category “Picked-for-you” was a new category added after the data had been extracted, menu items under this category were manually coded during the nutritional analysis. Value bundles such as meal deals, and family deals were coded manually during the nutritional analysis using the menu items’ names and descriptions. Menu items with value bundles (i.e., meal deals and family deals) were expected to increase the median price of the menu items due to their higher costs, and thus were excluded from the price analysis. Additionally, promotional offers such as “Buy 1, get 1 free”, “Free with $20 purchase (add to cart)”, etc., categories displayed by some food outlets were also manually entered under ‘Special Promotions’ during the nutritional analysis as they were not present during the original web scraping process. Special promotions were excluded from price analysis. Menu items that were under the “catering” or “party” categories or had similar terms in their descriptions or names were coded under the ‘Catering and Party Packs’ category. Some menu items were categorized under both ‘catering and party packs’ and ‘family deals’. An example of this is a menu item under the Uber Eats category ‘Family Meals’ and named ‘Party Packs’. Table 1 provides a summary of the definitions of these study outcomes.

### 2.4. Data Analysis

All data was collated on Microsoft Excel (Version 16.56, Microsoft Corporation, Redmond, Washington, DC, USA). Food and beverage categories that were less than 10% of all menu items were grouped into four categories: Other Food (discretionary), Other Food (FFG), Other Beverage (discretionary), and Other Beverage (FFG). To analyse the nutritional quality and the marketing attributes of all menu items, descriptive statistics such as proportions were calculated (Table 2). Categorical variables (nutritional quality, popularity, value bundles, photo, special promotions, nutritional information, and dietary labelling) were summarized using frequencies and proportions. All inferential statistical analysis were performed using SPSS Statistics Version 27 (IBM, Armonk, New York, NY, USA). Chi-squared tests with Bonferroni multiple comparisons correction and odds ratios were used for categorical variables to identify significant differences between (i) discretionary and FFG menu items (Table 3 and Table 4) and (ii) popularized and regular menu items (Table 4) (iii) independent outlets and franchise outlets. The variable ‘price’ was summarized as medians and interquartile intervals. Finally, Kruskal–Wallis tests with multiple comparisons corrections were used for continuous variable (price) to identify significant differences between (i) popularized menu items and regular menu items (Table 5) and (ii) comparable discretionary and FFG food and beverage categories (Table 5).

## 3. Results

### 3.1. Selection of Menu Items

A total of 29,764 menu items were available from complete menus of 354 unique popular independent and franchise restaurants and takeaway outlets available on Uber Eats in NZ (Figure 1). Of all menu items, 141 undetermined and non-consumable menu items were excluded. A total of 29,623 menu items, 3746 of which were combination menu items (that were analysed separately), were available for analyses of nutritional quality and marketing attribute, excluding price. Out of 25,877 menu items, 1571 were popularized menu items on Uber Eats. Following further exclusion of 513 catering and party packs, meal deals, family deals, and menu items with price unavailable, 25,364 menu items remained for price analysis, out of which 1518 were a popularized category.

Out of 29,764 menu items, 3748 (12.6%) were combination menu items (Figure 1). Of the 354 restaurants and takeaway outlets that were analysed, 86.2% (305/354) included combination menu items. After excluding 2 undetermined menu items, 3746 combination menu items remained for nutritional analysis and all marketing attributes, excluding price. Out of 3746 combination menu items, 442 were popularized by Uber Eats. As combination menu items consist of more than one food/beverage category, analysis for nutritional quality and marketing attributes was conducted by including all menu items which contained the food/beverage categories of interest. Following further exclusion of 2212 catering and party packs, meal deals, and family deals, with 6 menu items consisting of both catering and party packs and family deals, 1545 menu items were available for price analysis, out of which 194 were popularized menu items.

### 3.2. Nutritional Quality and Marketing Attributes

#### 3.2.1. Nutritional Quality and Popularized Menu Items

Table 2 shows the proportions of each food and beverage category for all menu items. Most of all menu items were discretionary (73.3%, 18,955/25,877). The discretionary cereal-based mixed meal category was the largest category within complete menus (36.4%, 9419/25,877) (Table 2). This category included burgers, pizzas, sandwiches, wraps, pasta, entrées, and sides. The second largest category was discretionary meat or alternative-based mixed meals (8.9%, 2308/25,877). This category included menu items such as deep-fried meat and seafood meals, processed meats, ribs, meat, or seafood meals with an excessive amount of sauce added as shown in the menu photo (if available), or meat/seafood curries including discretionary ingredients such as butter and cream (e.g., butter chicken) or other coconut-based or cream-based curries, including paneer or tofu.

**Table 2 nutrients-14-04567-t002:** The proportion of food and beverage categories in complete menus (*N* = 25,877) from 354 local independent and franchise restaurants and takeaway outlets in descending order. Excludes combination menu item menu items.

Type of Category	Food Categories	*n*	%
**Discretionary**	Cereal-based mixed meal	9419	36.4
	Meat or alternative based mixed meal	2308	8.9
	Sugar-Sweetened Beverages	1673	6.5
	Baked goods/Desserts (homemade or similar)	1001	3.9
	Other Beverage ^b^	979	3.8
	Vegetable-based mixed meal	763	2.9
	Discretionary Milk Based Beverages	698	2.7
	Savoury Sauces, Condiments and Spreads	670	2.6
	Iced confectionary and dairy-based desserts	617	2.4
	Fried Potato (or similar)	445	1.7
	Other Food ^a^	382	1.5
	Total Discretionary	18,955	73.3
**Five Food Groups (FFG)**	Cereal-based mixed meal	2366	9.1
	Meat or alternative based mixed meal	1459	5.6
	Vegetable-based mixed meal	889	3.4
	Other Beverage ^d^	811	3.1
	Other Food ^c^	533	2.1
	Juice	482	1.9
	Water	382	1.5
	Total FFG	6922	26.7
	Total	25,877	

^a^ Confectionery, Discretionary snack food (Savoury)—Packaged, Discretionary snack food (Sweet)—Packaged, Other snack food (other), Processed Meats, ^b^ Alcohol, Energy Drinks, Non-Sugar Sweetened Beverages, Rehydration Beverages (Electrolytes), Water Based Flavoured Beverage—sugar not determined, ^c^ Breads and Cereals, Dairy and alternatives, Fats/Oils, Fruit, Legumes, Meat and Alternatives, Soup, Vegetables (Other), ^d^ Body Building and Performance Beverages, Coffee, Milk/Milk Alternatives, Milk/Milk Alternative Based Beverages, Tea.

Table 3 shows the proportion of discretionary and FFG menu items within complete menus and each marketing characteristic. Popularized menu items comprised 6.1% (1571/25,877) of complete menus and the majority of the popularized menu items were significantly discretionary (83.8%, 1317/1571) (Table 3). Furthermore, a discretionary menu item was more likely (OR: 2.0, 95% CI 1.7–2.2) to be popularized compared to a FFG menu item. The discretionary cereal-based mixed meal category was the largest category from the popularized menu items (48.9%, 768/1571). The second-largest category for popularized menu items was meat or alternative-based mixed meals (19.7%, 310/1571).

**Table 3 nutrients-14-04567-t003:** The proportion of discretionary categories compared against Five Good Group (FFG) categories within marketing attributes for complete menus ^1^. Excludes combination menu item menu items.

Characteristic	Discretionary (%)	Five Food Group (%)	Total	Odds Ratio (95% CI)
Popularized Menu Items	1317 (83.8)	254 (16.2)	1571	2.0 (1.7–2.2) **
Photo	13,591 (76.7)	4132 (23.3)	17,723	1.7 (1.6–1.8) **
Value Bundle	363 (92.6)	29 (7.4)	392	4.6 (3.2–6.8) **
Special Promotions	265 (74.4)	91 (25.6)	356	1.1 (0.8–1.4)
Total Menu Items	18,955 (73.3)	6922 (26.7)	25,877	

^1^ The odds ratio was calculated for discretionary categories compared against FFG categories. The percentages are within each marketing attribute. ** Statistically significant *p* < 0.001. Abbreviation: CI = confidence interval.

#### 3.2.2. Nutritional Quality and Combination Popularized Menu Items

The proportion of each food and beverage category for popularized and regular menu items within combination menu items is provided in Appendix A. Since combination menu items contain more than one food/beverage item, analysis of the nutritional quality and marketing attributes included all menu items which contained the food/beverage category of interest. The majority (90.0%, 1176/1306) of popularized combination menu items and 81.1% (6941/8562) of remaining combination menu items contained at least one discretionary food/beverage category (Appendix A). Fried potato (or similar) was the largest category for both popularized unique combination menu items (65.4%, 289/442) and for the remaining combination menu item (53.0%, 1752/3304). This category mainly included potato/kumara fries or wedges that were one of the food items in the meal deal or combination menu item. Discretionary cereal-based mixed meals were the second largest food category for both popularized unique combination menu items (51.1%, 226/442) and the remaining combination menu item (47.7%, 1577/3304). This included combo meals (i.e., meal deals) or combination menu items that contained discretionary cereal-based mixed meal items such as burgers, wraps, discretionary breads (i.e., butter naan, garlic bread, etc.), and pizzas.

Popularized unique combination menu item menu items comprised 11.8% (442/3746) of total unique combination menu items and the majority of the popularized menu items were discretionary (90.0%, 1176/1306). There were no significant differences between all four outlet categories (independent takeaway, independent restaurant, franchise takeaway, franchise restaurant) in terms of nutritional quality for all menu items.

#### 3.2.3. Photos

Within complete menus, 68.5% (17,723/25,877) of the menu items were accompanied by a photo (Table 4). A higher proportion of discretionary menu items (71.7%, 13,591/18,955) had photos compared to FFG menu items (59.7%, 4132/6922) (*p* < 0.001) (Table 4). Discretionary menu items were 1.7 times more likely (OR: 1.7, 95% CI 1.6–1.8) to include a photo as compared to the FFG menu items (Table 3). Among the popularized menus, 78.5% (1234/1571) were accompanied by a photo and were also more likely to have a photo compared to all menu items (*p* < 0.001). A higher proportion of discretionary popularized menu items (82.7%, 1089/1317) had photos compared to FFG popularized menu items (57.1%, 145/254) (Table 4).

**Table 4 nutrients-14-04567-t004:** Prevalence of photos, value bundles and special promotions for all menu items (*N* = 25,877) and popularized menu items (*n* = 1571) of 354 independent and franchise restaurants and takeaway outlets ^1^. All menu items include popularized menu items and remaining regular menu items.

Food & Beverage Group	Food & Beverage Category	Marketing Attributes	All Menu Items		Popularized Menu Items	
			*n*	%	*n*	%
**Food** **(Discretionary)**	Cereal-based mixed meal	Photo	7193	76.4	671	87.4 **
		Value Bundle	105	1.1	3	0.4 *
		Special Promotions	151	1.6	10	1.3
	Meat or alternative-based mixed meal	Photo	993	43.0	236	76.1 **
		Value Bundle	69	3.0	37	11.9 **
		Special Promotions	13	0.6	1	0.3
	Savoury Sauces, Condiments and Spreads	Photo	260	38.8	2	50.0
		Value Bundle	0	0	0	0
		Special Promotions	0	0	0	0
	Fried Potato (or similar)	Photo	316	71.0	30	65.2
		Value Bundle	1	0.2	1	2.2 *
		Special Promotions	3	0.7	0	0
**Food** **(Discretionary)**	Baked goods/ desserts (homemade or similar)	Photo	894	89.3	49	98.0 *
		Value Bundle	181	18.1	6	12.0
		Special Promotions	30	3.0	0	0
	Iced confectionary and dairy-based desserts	Photo	565	91.6	21	87.5
		Value Bundle	2	0.3	0	0
		Special Promotions	2	0.3	0	0
	Vegetable-based mixed meal	Photo	386	50.6	14	31.1 *
		Value Bundle	5	0.7	0	0
		Special Promotions	13	1.7	1	2.2
	Other Food ^a^	Photo	331	86.6	5	100.0
		Value Bundle	0	0	0	0
		Special Promotions	1	0.3	0	0
**Beverage** **(Discretionary)**	Sugar-Sweetened Beverages	Photo	1208	72.2	8	72.7
		Value Bundle	0	0	0	0
		Special Promotions	22	1.3	0	0
	Other Beverage ^b^	Photo	843	86.1	10	90.9
		Value Bundle	0	0	0	0
		Special Promotions	8	0.8	0	0
	Milk Based Beverages	Photo	602	86.2	43	100.0 *
		Value Bundle	0	0	0	0
		Special Promotions	22	3.2	2	4.7
**Total** **Discretionary**		Photo	13,591	71.7 ^#^	1089	82.7
		Value Bundle	363	1.9 ^#^	47	3.6
		Special Promotions	265	1.4	14	1.1
**Food (FFG)**	Cereal-based mixed meal	Photo	1618	68.4	82	60.3
		Value Bundle	21	0.9	4	2.9
		Special Promotions	28	1.2	1	0.7
	Vegetable-based mixed meal	Photo	427	48.0	9	42.9
		Value Bundle	0	0	0	0
		Special Promotions	8	0.9	0	0
	Meat or alternative-based mixed meal	Photo	422	28.9	50	58.1 **
		Value Bundle	8	0.5	2	2.3 *
		Special Promotions	1	0.1	0	0
	Other Food ^c^	Photo	173	32.5	2	22.2
		Value Bundle	0	0	0	0
		Special Promotions	2	0.4	0	0
**Beverage (FFG)**	Water	Photo	329	86.1	0	0
		Value Bundle	0	0	0	0
		Special Promotions	8	2.1	0	0
	Other Beverage ^d^	Photo	763	94.1	2	100.0
		Value Bundle	0	0	0	0
		Special Promotions	32	3.9	0	0
	Juice	Photo	400	83.0	0	0
		Value Bundle	0	0	0	0
		Special Promotions	12	2.5	0	0
**Total FFG**		Photo	4132	59.7	145	57.1
		Value Bundle	29	0.4	6	2.4
		Special Promotions	91	0.5	1	0.4
**Total**		Photo	17,723	68.5	1234	78.5 **
		Value Bundle	392	1.5	53	3.4 **
		Special Promotions	356	1.4	15	1.0

^a^ Confectionery, Discretionary snack food (Savoury)—Packaged, Discretionary snack food (Sweet) Packaged, Other snack food (other), Processed Meats. ^b^ Alcohol, Energy Drinks, Non-Sugar Sweetened Beverages, Rehydration Beverages (Electrolytes), Water Based Flavoured Beverage—sugar not determined. ^c^ Breads and Cereals, Dairy and alternatives, Fats/Oils, Fruit, Legumes, Meat and Alternatives, Soup, Vegetables, Vegetables (Other). ^d^ Body Building and Performance Beverages, Coffee, Milk/Milk Alternatives, Milk/Milk Alternative Based Beverages, Tea. ^1^ Percentages are within each Food & Beverage Category where displayed, otherwise within the Total. ** *p* < 0.001 compared to all menu items and their FFG or Discretionary counterpart. * *p* < 0.01 compared to all menu items and their FFG or Discretionary counterpart. ^#^
*p* < 0.001 compared to Total FFG (All Menu Items). Chi-squared tests with Bonferroni multiple comparisons correction were used to generate *p* values.

#### 3.2.4. Value Bundles

Within all menu items, 1.5% (392/25,877) were a value bundle (Table 4). A higher proportion of discretionary menu items (1.9%, 363/18,955) were offered as a value bundle compared to FFG menu items (0.4%, 29/6922) (*p* < 0.001) (Table 4). Discretionary menu items were more likely (OR: 4.6, 95% CI 3.2–6.8) to be offered as a value bundle compared to a FFG menu item (Table 3). Baked goods/desserts (homemade or similar) made up 46% (181/392) of all value bundles, the largest category for this marketing attribute (Table 4). There was a significant difference in the number of value bundles within the popularized menu items compared to regular menu items (*p* < 0.001). Like complete menus, a higher proportion of discretionary popularized menu items (3.6%, 47/1317) were offered as a value bundle compared to FFG popularized menu items (2.4%, 6/254) (*p* < 0.001) (Table 4).

#### 3.2.5. Special Promotions

Within complete menus, 1.4% (356/25,877) of the menu items included special promotions (Table 4). A higher proportion of discretionary menu items (74.4%, 265/356) included special promotions compared to FFG menu items (25.6%, 91/356) (Table 4). However, there was no significant difference in the number of special promotions within the discretionary menu items compared to FFG menu items (*p* = 0.610). Similarly, no significant difference was found in the number of special promotions within the popularized menu items compared to all menu items (*p* = 0.139).

#### 3.2.6. Price

Table 5 shows the median prices (in NZD) of the (i) popularized and (ii) regular menu items, excluding catering and party packs and value bundles. The median price of the popularized menu items was significantly higher than regular menu items for savoury sauces, condiments, and spreads (*p* = 0.019), fried potato (or similar) (*p* < 0.001), baked goods/desserts (homemade or similar) (*p* < 0.001), discretionary vegetable-based mixed meals (*p* = 0.001), iced confectionery and dairy-based desserts (*p* < 0.001), FFG cereal-based mixed meals (*p* < 0.001), FFG vegetable-based mixed meals (*p* < 0.001) and discretionary other beverages (*p* = 0.003). However, the median price of the popularized menu items for FFG meat or alternative-based mixed meals, was significantly less than the regular menu items (*p* = 0.004). Additionally, the median price of the popularized menu items for discretionary cereal-based mixed meals was the same as regular menu items, hence no significant difference was found (*p* = 0.128).

Figure 2 compares the median price between categories with discretionary and FFG counterparts for all menu items. All menu items include both regular menu items (unpopularized) and popularized menu items which have been separately identified in Table 5. The median price for discretionary cereal-based mixed meals ($12.60) was higher than its FFG counterpart ($11.60). However, this price difference between discretionary and FFG cereal-based mixed meals was not significant (*p* = 0.607). On the other hand, the median price of both discretionary meat or alternative-based mixed meals ($16.00) and discretionary vegetable-based mixed meals ($12.20) was lower than their FFG counterpart ($19.00, *p* < 0.001 and $14.00, *p* < 0.001 respectively).

**Table 5 nutrients-14-04567-t005:** Median price (in NZD) of popularized and regular menu items for each food or beverage category. Regular menu items exclude popularized menu items. Catering and party packs, value bundles, combination menu items and menu items with price unavailable were excluded (*N* = 25,364).

Food or Beverage Group	Food or Beverage Category	Popularized Menu Items		Regular Menu Items				*p*-Value
		Median Price ($)	Q1	Q3	Median Price ($)	Q1	Q3	
**Food** **(Discretionary)**	Cereal-based mixed meal	12.60	10.20	16.00	12.60	8.95	15.90	0.128
	Meat or alternative based mixed meal	15.98	8.20	18.97	16.00	10.00	19.90	0.865
	Savoury Sauces, Condiments and Spreads	5.00	3.13	8.00	1.50	0.50	4.00	0.019 *
	Fried Potato (or similar)	6.99	5.25	9.90	4.99	3.83	7.43	0.000 **
	Baked goods/desserts (homemade or similar)	9.95	5.75	17.40	5.20	2.20	7.90	0.000 **
	Vegetable-based mixed meal	16.00	13.80	19.50	12.20	8.40	17.00	0.000 *
	Iced confectionary and dairy-based desserts	12.99	6.90	15.24	6.10	4.90	12.99	0.000 **
	Other Food ^a^	6.45	6.05	7.23	6.75	4.45	8.10	0.909
**Food (FFG)**	Cereal-based mixed meal	16.99	13.00	18.50	11.60	9.80	16.99	0.000 **
	Vegetable-based mixed meal	19.00	15.00	18.50	14.00	10.40	18.00	0.000 **
	Meat or alternative based mixed meal	17.95	15.99	21.25	19.00	16.90	21.90	0.004 *
**Food (FFG)**	Other Food ^c^	5.90	2.75	7.45	4.00	2.59	8.00	0.639
**Beverage** **(Discretionary)**	Sugar Sweetened Beverages	5.65	3.00	6.45	4.60	4.00	5.40	0.103
	Other Beverage ^b^	6.35	5.70	6.90	5.40	4.40	5.80	0.003 *
	Discretionary Milk Based Beverages	6.50	4.30	8.00	5.76	5.10	7.25	0.265
**Beverage (FFG)**	Water	-	-	-	4.15	3.60	5.00	-
	Other Beverage ^d^	7.33	5.65	-	5.00	4.50	6.00	0.075
	Juice	-	-	-	4.80	4.10	5.00	-

^a^ Confectionery, Discretionary snack food (Savoury)—Packaged, Discretionary snack food (Sweet) Packaged, Other snack food (other), Processed Meats. ^b^ Alcohol, Energy Drinks, Non-Sugar Sweetened Beverages, Rehydration Beverages (Electrolytes), Water Based Flavoured Beverage—sugar not determined. ^c^ Breads and Cereals, Dairy and alternatives, Fats/Oils, Fruit, Legumes, Meat and Alternatives, Soup, Vegetables, Vegetables (Other). ^d^ Body Building and Performance Beverages, Coffee, Milk/Milk Alternatives, Milk/Milk Alternative Based Beverages, Tea. Q = Quartile. * *p* < 0.05. ** *p* < 0.001. Kruskal–Wallis tests with multiple comparisons corrections were used to generate the *p-*values.

#### 3.2.7. Nutritional Information and Dietary Labelling

Nutritional information was available for 19.7% (5095/25,877) of all menu items and only energy (kJ/kcal) values were provided for most of these menu items. Out of 5096 menu items with nutritional information, 99% (5061/5095) were provided for Subway takeaway outlet menu items which included a link to their website for all nutritional information, including nutrition information panel, ingredients, and allergens for their complete menu. Dietary labelling was found for 714 menu items which comprised mostly of gluten-free, vegan, and vegetarian labels. Nutrition information was absent from all outlets except for one franchise outlet, Subway.

## 4. Discussion

This study is one arm of a multi-national study, and to our knowledge, the first to assess the nutritional quality and marketing attributes of menu items of popular independent and franchise restaurants and takeaway outlets on a market leading OFD platform in NZ. Discretionary food and beverages made up majority of the menus and compared to FFG menu items, were more likely to be in the popularized section, accompanied by a photo, offered as a value bundle, and included in special promotions. Nutritional and dietary labelling was also excluded from a large proportion of menu items from the restaurants and takeaway outlets. Additionally, discretionary mixed meals were less expensive than their FFG counterparts.

Approximately three-quarters (73.3%) of all menu items from popular independent and franchise restaurants and takeaway outlets were discretionary and were more likely to be popularized and marketed to the consumers on the OFD platform. Almost 9 out of 10 of the popular menu items were discretionary in Sydney and Auckland [19]. Likewise, 9 out of 10 of the popularized combination menu items included discretionary food/beverage items, hence classified as unhealthy. Findings from studies conducted by Poelman et al. [41] and Jaworowska et al. [42,43], also align with the results of our study, that a large proportion of menu items offered on OFD platforms are of poor nutritional quality. Regardless of outlet type (franchise or independent, restaurant or takeaway), there were no significant differences in the proportion of discretionary foods available. Discretionary menu items were 2 times more likely (OR: 2.0, 95% CI 1.7–2.2) to be popularized which means unhealthy food and beverages are more visible to consumers on the OFD platform when a food outlet is selected, compared to healthier menu items.

There is a possibility that the COVID-19 pandemic accelerated the use of OFD platforms [26,44,45]. Since all food outlets were forced to shut during the initial Alert Level 4 lockdown in NZ, including OFD services, consumers had no access to restaurants and takeaway foods during those seven weeks [46]. Once these restrictions were slightly eased under Level 3, permitting takeaway outlets and OFD platforms to resume, there was a significant increase in takeaway sales, including online, hence a predicted increase in consumption of unhealthy foods and beverages [47,48]. These reports also align with Uber’s yearly report which disclosed that their delivery bookings grew 113% in the second quarter of 2020 and revenue increased by 103% in August 2020 as compared to the previous year [49]. Uber also reported seeing a 30% increase in customers signing up for the service in the same year [50]. Furthermore, there was a shift in dietary patterns towards more discretionary foods and beverages, and a selection of unhealthier food options due to pandemic-related anxiety in NZ [46,51]. This raise concerns as an increase in consumption of energy-dense and nutrient poor foods further increases health risks such as obesity and NCDs [14]. Furthermore, the WHO Obesity Report 2022 has also reported that individuals carrying excess bodyweight and/or those with NCDs have shown to have an increased risk of mortality from COVID-19 [52].

Approximately half of the popularized menu items in NZ were discretionary cereal-based mixed meals such as burgers and pizza. For combination menu items, discretionary cereal-based mixed meals were included in more than half of the popularized combination menu items, however fried potato (or similar) was the most dominant food category and was included in approximately 7 out of 10 of the popularized combination menu items (Appendix A). Another cross-sectional study conducted across three international cities also found that pizzas and burgers were the most common food options to be promoted and marketed on the platform [41]. Discretionary menu items are likely to be frequently purchased and remain under the popularized category for the individual food outlets, if the Uber Eats algorithm for popularized menu items is based on sales [53].

Discretionary menu items were more likely to be accompanied by a photo, offered as a value bundle, and have special promotions as compared to FFG menu items on Uber Eats. Uber Eats is therefore not only dominated by discretionary foods and beverages, but discretionary menu items are also largely subjected to the use of marketing attributes. This was also the case in Sydney, Australia [22]. Additionally, Uber Eats also reported a massive surge in “family meals” with bundles in 2021 for Australia and NZ, as more people were spending time at home [53]. Discretionary menu items were approximately five times more likely to be offered as a value bundle as compared to an FFG menu item. This raises public health concerns as value bundles increase the energy content of the meal without adding any beneficial nutrients [54]. Similarly, a 2020 study in Brazil reported that unhealthier foods were advertised more frequently and were marketed more predominantly using discounts, free deliveries, and combos, compared to healthier foods [28]. Another Australian study reported that marketing attributes such as price, appealing food images, and value for money food items significantly influence the dietary preferences of young people [16].

Images are strong influencers of dietary choice [16]. Popular OFD platforms have also reported that appealing photography of the menu items on the digital platform is a key factor in increasing consumer’s appetite and boosting sales [55]. A 2017 study in the US also reported that presenting attractive pictures of menu items increases consumers’ attitudes, purchasing intentions, and willingness to pay, therefore increasing sales and profit [56]. Despite multiple studies reporting that food and beverages with poor nutritional quality are more likely to have images, further research is required regarding how these images influence consumers’ purchasing and dietary behaviours on OFD platforms.

Special promotions and offers were another marketing characteristic analysed which was unique to this study. Special promotions are price-based promotions that provide consumers with discounts if they purchased food of up to a certain amount (i.e., ‘free delivery’ if cart total is up to $30). Discretionary menu items were more likely to have special promotions than FFG menu items. More recently, Uber has also introduced another feature for all its’ services called ‘Uber Pass’, a monthly subscription providing discounts and offers, including free delivery on food and groceries [57,58]. Since young adults are the largest and more frequent users of Uber Eats [20], they are also highly likely to have subscribed to this service considering expensive delivery charges on some food delivery. However, this data is not made publicly available.

According to previous research, consumers’ choice of food outlets on OFD platforms have shown to be influenced by average price when comparing within the same cuisine [59]. Considering that young individuals are the largest users of OFD platforms they are more likely to be price-sensitive given the limited budgets of full or part-time students [16]. Menu items from two of the three discretionary mixed meal categories were significantly less expensive than their comparable FFG category. The FFG cereal-based mixed meal was the only category where meals were cheaper than the discretionary counterparts. Similarly, the Australian study found that two out of three FFG mixed meal categories were significantly less expensive than their discretionary counterparts [22]. However, they also reported that despite the significant difference between the prices of healthy and unhealthy mixed meals, it is undetermined whether a difference of $1-$2 would make any difference in purchasing and it is unknown whether consumers choose healthier meals from a price perspective [22]. The common public perception is that healthy food is more expensive than unhealthy food [60,61], even though a study in NZ has shown that is not the case [62]. Moreover, as taste has also been shown to be an important characteristic in influencing dietary behaviours, there are high chances that the palatability of the food is more important to the consumers, especially when the price difference is not significant [16]. Interestingly, discretionary food/beverage categories that were popularized on Uber Eats, were significantly more expensive than the same food/beverage categories not popularized. Popular menu items are likely to be the ones that are more frequently sold [53].

The meat or alternative-based mixed meals were overall highly priced than both the cereal and vegetable-based mixed meals. This result was expected as meat is usually more expensive than vegetables [63].

Despite menu labelling being reported as an effective method of allowing consumers to make more informed choices when selecting food prepared away from home [64], not many food outlets have been shown to offer nutritional information on OFD platforms. The Australian study also reported only 0.2% of all menu items offering menu kilojoule labelling on restaurants/takeaway outlets on Uber Eats [22]. Most chain franchise outlets in Australia are subject to mandatory menu kilojoule labelling policy [5]. Nutrition information was basically absent from all outlets in NZ except for one franchise outlet, Subway, therefore, indicating how challenging it is for consumers to choose healthier menu items on OFD platforms such as Uber Eats. Mandatory kilojoule labelling is a possible policy intervention that could also be implemented in New Zealand

This current study in NZ provides us with an understanding of the digital food environment created by the leading OFD platform. The comprehensive classification system of 38 food and beverage categories used for nutritional analysis also enabled a deeper understanding of the nutritional quality of menu items available on the digital platform. Additionally, separate nutritional coding of the combination meals was a unique aspect of this study.

However, there are several limitations to this study that have arisen due to the nature and process of the research. Although the cross-sectional study design can be used to prove assumptions, the digital food environment is likely to evolve rapidly and frequently which can mean changes in results are highly likely if the same research were to be conducted again. Most of the restaurants and takeaway outlets included in the study were established in Auckland and so cannot be generalized to all of NZ. A further limitation was that we only assessed the 10 most popular independent and franchise restaurants and takeaway outlets identified in each suburb. Therefore, there is a possibility that some healthy outlets have been masked by the popular, predominantly unhealthy food outlets. Furthermore, since this study only examined the market leading OFD platform in NZ (i.e., Uber Eats), we may have excluded other OFD platforms with widespread usages such as ‘Delivereasy’ and ‘Menulog’ and any outlets exclusive to them. We did not explore whether there were any changes in the nutritional quality, marketing, or price of the menu items before and after the COVID-19 lockdown.

## 5. Conclusions

Our findings suggest that the menus from popular local independent and franchise restaurants and takeaway outlets on Uber Eats in NZ are dominated by menu items that are predominantly unhealthy, with a higher energy content and poor nutritional quality. These menu items appear to be largely available and exposed to the users through the extensive use of marketing attributes. They are also more likely to appeal to price-sensitive consumers. Special promotions were an additional marketing characteristic that was unique to this NZ study. The addition of this price-based promotional strategy highlights how rapidly OFD platforms invent new techniques to appeal to its users’ purchasing behaviours. In this current digitally led world, it is no surprise that OFD services are growing popularity, with a further acceleration since the start of the global pandemic. These services effectively meet the consumer’s demand for convenience, bringing their favourite cuisines and cravings at their fingertips by increasing availability and accessibility through such platforms, further promoting a sedentary lifestyle. The results of this study also align with the other arm of this study conducted in Australia, regarding the healthiness of the digital food environment. Furthermore, multiple studies over the years have raised concerns regarding the nutritional quality of foods offered on OFD platforms, and meals prepared outside of home being higher in energy, with an undesirable nutritional profile in general. Frequent consumption of fast-foods and takeaways with poor nutritional quality is a significant contributor to the alarming increase in prevalence of obesity and non-communicable diseases, especially for the young population. The overwhelming availability and promotion of discretionary menu items on Uber Eats are therefore likely to influence the nutritional quality of the food choices users make. However, we neither examined the consumption nor the sales of the menu items on Uber Eats, therefore, these associations of the unhealthy digital food environment with the health consequences remain hypothetical. Hence, further research is strongly advocated and the need for menu kilojoule labelling policies or similar public health interventions to enable consumers to use convenience for healthier options.

## Figures and Tables

**Figure 1 nutrients-14-04567-f001:**
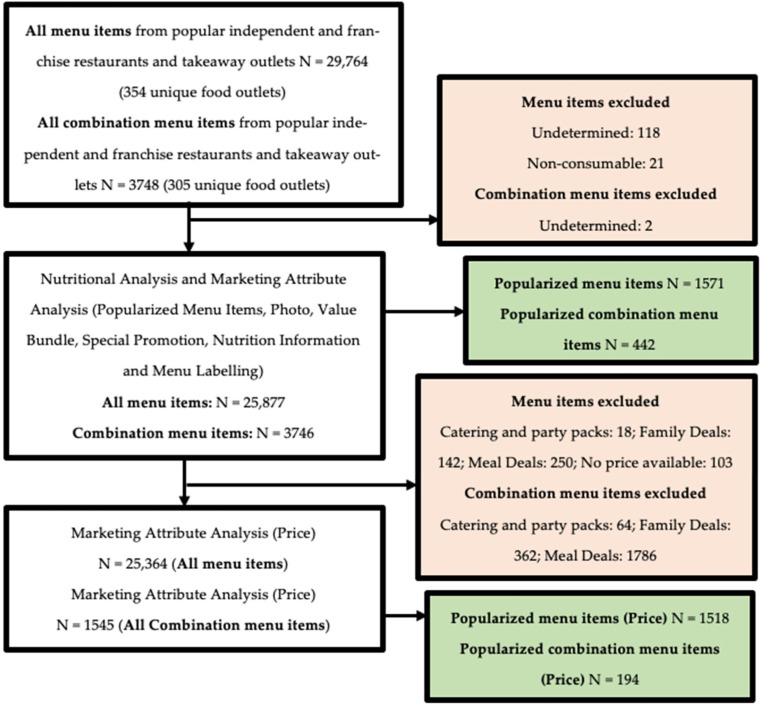
Flow diagram outlining the inclusion of menu items in each analysis.

**Figure 2 nutrients-14-04567-f002:**
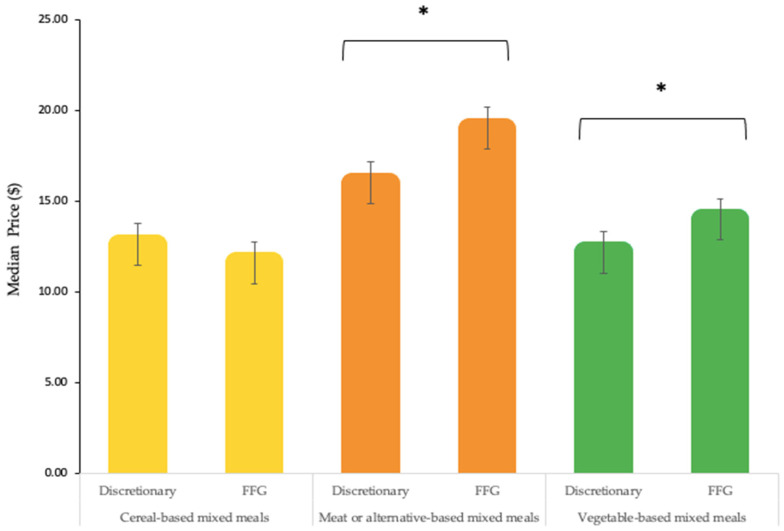
Median price (in NZD) of all discretionary and Five Food Group (FFG) mixed meal menu items excluding catering and party packs, value bundles, combination menu items and menu items with price unavailable. All menu items include regular menu items and popularized menu items. Error bars display interquartile interval, * *p* < 0.001.

**Table 1 nutrients-14-04567-t001:** Summary of definitions and derivations of data extracted from complete menus of each independent and franchise food outlet on Uber Eats and study outcomes.

**Data Extracted**	**Definition**
Menu item name [22]	The name given to menu items as displayed on the food outlet’s webpage.
Menu item description [22]	The description of the menu items available under the menu item name on the food outlet’s webpage. Descriptions are not provided for all menu items.
Uber eats category [22]	The food category within which menu items are grouped on the food outlet’s webpage (e.g., Desserts, Sides, Mains, etc.).
Catering and party packs [22]	Any menu item with terminologies like “catering”, “party”, or similar in either the Uber Eats category or the menu item name. The menu items under this category were assumed to cater 10 people or more.
Uber Eats category duplicate [22]	Menu items that were repeated in more than one Uber Eats categories. These menu items were listed both as “Most Popular” or ‘’Picked For You’’ and as another Uber Eats category (e.g., Kumara Fries listed under “Picked For You” and “Sides” categories).
Meal deal [22]	Any menu item comprising of more than one food item which could be individually purchased from the food outlets (e.g., pizza with drink and dessert). The food items under this category were available to purchase at a lower price compared to purchasing the food items individually. This was determined using the description provided of the menu item in the context of the complete menu of the food outlet.
Family deal [22]	Any menu item designed to serve greater than one person and assumed to cater less than ten people. The menu items under this category included the terms “for two”, “for three”, “family”, or similar in the Uber Eats category, menu item name or description.
**Study Outcomes**	**Definition**
Discretionary food or beverage	According to the Australian Dietary Guidelines [23], foods and beverages under this category are defined as items high in saturated fat, added sugars, salt and low in dietary fibre. Internationally, they are also referred to as junk food or non-core.
Five Food Group (FFG) food or beverage	According to the Australian Dietary Guidelines, foods and beverages under this category are expected to include food(s) or combination of foods from the vegetables and legumes/beans, fruit, grain (cereal) foods, mostly wholegrain and/or high cereal fibre varieties; lean meats and poultry, fish, eggs, tofu, nuts and seeds and legumes/beans; and milk, yoghurt, cheese and/or alternatives, mostly reduced fat [23].
Most popular menu items [22]	Menu items listed under the “Most Popular” category on the food outlet’s webpage on Uber Eats. Menu items under this category are generally located right at the top of the food outlet’s Uber Eats webpage as well as on the Uber Eats app, making them easily visible. The remaining menu items are known are regular menu items.
Picked for you menu items	Assumption made that it is like most popular menu items.
Popularized menu items	This is a collective term for the menu items listed as either “Most Popular” or “Picked for you” in the Uber Eats category.
Combination menu items	Menu items containing more than one food/beverage item. Each food/beverage item is analysed individually for nutritional quality.
Value bundles	Menu items under this category are included under both meal deals and family deals category on Uber Eats.
Special Promotions	Menu items that include promotions such as “Buy 1, get 1 free”, “one free drink”, “$0 delivery fee (spend $20)”, etc. Additional promotional slogans presented with menu items This was determined using the menu item description or food outlet webpage.
Photo [22]	The photo available on the Uber Eats webpage accompanying the menu item name, description, and price. Photos are not available for all menu items.
Price ($) [22]	The amount, in NZ dollars menu items are sold for on the food outlet’s Uber Eats webpage.
Nutritional Information [22]	Any information provided on the food outlet’s Uber Eats webpage that quantifies any macronutrient(s) of a menu item (e.g., energy, fats, protein) or micronutrient(s) (e.g., sodium). Nutritional information is not provided for all menu items.
Dietary labelling [22]	Any menu item labelled with a special dietary requirement (e.g., vegetarian, vegan, gluten free). Religious dietary labelling (e.g., halal) and heat scale labelling (e.g., spicy) was excluded from these data. Dietary labelling is not available for all menu items.

## Data Availability

Publicly available datasets were analysed in this study. This data can be found here: https://www.ubereats.com/nz (accessed on 14 April 2021).

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
