# Peer review of "Unhealthy Food at Your Fingertips: Cross-Sectional Analysis of the Nutritional Quality of Restaurants and Takeaway Outlets on an Online Food Delivery Platform in New Zealand"

_nutrients, 2022, doi:10.3390/nu14214567_

Round 1

Reviewer 1 Report

After careful consideration, I fell that the manuscript entitled “Unhealthy Food at your Fingertips: Cross-Sectional Analysis of the Nutritional Quality of Takeaway Outlets on an Online Food Delivery Platform in New Zealand” has merit, but is not suitable for publication as it currently stands. Therefore, my decision is "Major Revision."

Here are my comments:

ABSTRACT
- lines 32-34: The authors state that: "Discretionary menu items were more likely to be categorized as most popular (OR: 2.0, 95% CI 1.7-2.2), accompanied by a photo (OR: 1.7, 95% CI 1.6-1.8), offered as a value 33 bundle (OR: 4.6, 95% CI 3.2-6.8) and include special promotions (OR: 1.1, 95% CI 0.8-1.4)". However in Special Promotions' OR, the 95% CI  includes the value 1 and therefore this difference is not significant at a 5% level of signinificance.

DATA ANALYSIS
- There is no description of the significance level (alpha) adopted in the tests performed. It should be included in this section.

- Line 247-248: The authors have stated that "The distribution of continuous variable (price) was assessed using histograms and measures of skewness and kurtosis. The variable 'price' was summarized as medians and interquartile intervals". However, no histograms were found in the paper, as well as the values of skewness and kurtosis.

- It is not clear how and where the Bonferroni correction and the multiple comparisons corrections (of Kruskall-wallis test) were implemented in the analysis. The tests were corrected for how many multiple tests? Describe this information further in the Results Section. One suggestion is that this information should be clear in the notes to the tables presenting the results of these tests.

RESULTS
-  lines 301-302: Which statistical test was performed that generated the presented p-value? This test should be described in the Data Analysis section. Suggestion: Since one group is complementary to the other, an alternative is to present the 95% CI of the percentage of one of the groups and consider significant (at a 5% significance level) if CI is greater than 50%.

- Table 3: The authors could include the category "other menu items" so as to make the sum agree with the "Total menu items".

- Table 3: To include, as it was done in Table 4, the markings (footnotes) of the significant differences

- Lines 331-332: What test was performed and what was the p-value to conclude that "There were no significant differences between independent and franchise outlets in terms of nutritional quality for all menu items"?

- Line 340: Which statistical test was performed that generated the presented p-value ?

- Table4: Why did the authors discriminate "* p<0.01" and "** p<0.001"?  What was the significance level of the adopted test?

- Table 4: The authors should describe in this footnote the test used that generated these p-values.

- Table 4: Since the p-values were presented in the text of the manuscript, a suggestion is that they (p-values) should be included in the Table (as is done in Table 5).

- Table 5: describe in the footnote the statistical test that generated the p-value. Furthermore, I believe that it is more correct here to consider the Mann-Whitney test instead of the Kruskall-Wallis test.

- Table 5: The authors should clarify how the correction for multiple testing was done (described in the Data analysis section). Are the p-values shown in the table the corrected values?

Author Response

Thank you for allowing us to submit a revised draft of my manuscript titled Unhealthy Food at your Fingertips: Cross-Sectional Analysis of the Nutritional Quality of Takeaway Outlets on an Online Food Delivery Platform in New Zealand to Nutrients. We appreciate the time and effort that you and the reviewers have dedicated to providing your valuable feedback on our manuscript. We are grateful to the reviewers for their insightful comments on our paper. We have been able to incorporate changes to reflect most of the suggestions provided by the reviewers. We have highlighted the changes within the manuscript.

Here is a point-by-point response to the reviewer 1’s comments and concerns.

Comments from Reviewer 1

  • Comment 1: lines 32-34: The authors state that: "Discretionary menu items were more likely to be categorized as most popular (OR: 2.0, 95% CI 1.7-2.2), accompanied by a photo (OR: 1.7, 95% CI 1.6-1.8), offered as a value 33 bundle (OR: 4.6, 95% CI 3.2-6.8) and include special promotions (OR: 1.1, 95% CI 0.8-1.4)". However, in Special Promotions' OR, the 95% CI includes the value 1 and therefore this difference is not significant at a 5% level of significance.
  • Response: Thank you for pointing this We agree with this comment. Therefore, we have deleted the odds ratio and confidence interval value for special promotions. This change can be found in Abstract, line 31.

  • Comment 2: Line 247-248: The authors have stated that "The distribution of continuous variable (price) was assessed using histograms and measures of skewness and kurtosis. The variable 'price' was summarized as medians and interquartile intervals". However, no histograms were found in the paper, as well as the values of skewness and kurtosis.
  • Response: The authors have decided to remove this sentence as the histogram and the values for skewness and kurtosis were not reported and visually represented in the manuscript for brevity purposes. This change can be found on page 6, paragraph 2.4., line 263-264.

  • Comment 3: It is not clear how and where the Bonferroni correction and the multiple comparisons corrections (of Kruskall-Wallis test) were implemented in the analysis. The tests were corrected for how many multiple tests? Describe this information further in the Results Section. One suggestion is that this information should be clear in the notes to the tables presenting the results of these tests.
  • Response: Thank you for this suggestion. The authors decided that it is more appropriate to clarify all the data analyses conducted in the data analysis paragraph. We have made the appropriate modifications to the data analysis paragraph which can be found on page 6, paragraph 2.4., lines 260-267.

  • Comment 4: Which statistical test was performed that generated the presented p-value? This test should be described in the Data Analysis section. Suggestion: Since one group is complementary to the other, an alternative is to present the 95% CI of the percentage of one of the groups and consider significant (at a 5% significance level) if CI is greater than 50%.
  • Response: Thank you for this suggestion, the authors have indicated in the data analysis section, what statistical tests were performed for each table in the results section. We have made the appropriate modifications to the data analysis section which can be found on page 6, paragraph 2.4., lines 260-267.

  • Comment 5: Table 3: The authors could include the category "other menu items" to make the sum agree with the "Total menu items".
  • Response: Thank you for this suggestion. However, the authors decided not to add “other menu items” as each row (individual marketing characteristics) adds up to the “total” menu items.

  • Comment 6: Table 3: To include, as it was done in Table 4, the markings (footnotes) of the significant differences
  • Response: Thank you for this suggestion. We have added the markings and footnotes of the significant differences.

  • Comment 7: Lines 331-332: What test was performed and what was the p-value to conclude that "There were no significant differences between independent and franchise outlets in terms of nutritional quality for all menu items"?
  • Response: Chi-squared tests with Bonferroni multiple comparisons correction and odds ratios were used for categorical variables to identify significant differences and this has been clarified in the data analysis section which can be found on page 6, paragraph 2.4., lines 260-267

  • Comment 8: Line 340: Which statistical test was performed that generated the presented p-value ?
  • Response: The authors have indicated in the data analysis section, what tables each statistical test refers to. We have made the appropriate modifications to the data analysis paragraph which can be found on page 6, paragraph 2.4., lines 260-267.

  • Comment 9: Table4: Why did the authors discriminate "* p<0.01" and "** p<0.001"?  What was the significance level of the adopted test?
  • Response: This study is part of a multi-city study conducted in Australia and New Zealand. The research team optimised all stages of our research to minimise sources of uncertainty. When presenting p values, we found it helpful to use the asterisk rating system and the p value: p < 0.01* p < 0.001**. We followed the convention in other published research in the area and referred to statistically significant as p < 0.01 and statistically highly significant as p < 0.001 and therefore discriminated between the two using asterisk.

  • Comment 10: Table 4: The authors should describe in this footnote the test used that generated these p-values.
  • Response: Thank you for this suggestion. The authors have added to the footnotes of relevant tables throughout the manuscript, tests used to generate the p-

  • Comment 11: Table 4: Since the p-values were presented in the text of the manuscript, a suggestion is that they (p-values) should be included in the Table (as is done in Table 5).
  • Response: Thank you for this suggestion. However, the authors decided to only mark the significant p-values using asterisk in Table 4 as it is consistent with the format of the tables with the Australian study (Hunger for Home Delivery).

Wang, C.; Korai, A.; Jia, S.S.; Allman-Farinelli, M.; Chan, V.; Roy, R.; Raeside, R.; Phongsavan, P.; Redfern, J.; Gibson, A.A.; Partridge, S.R. Hunger for Home Delivery: Cross-Sectional Analysis of the Nutritional Quality of Complete Menus on an Online Food Delivery Platform in Australia. Nutrients 2021, 13, 905. https://doi.org/10.3390/nu13030905

  • Comment 12: Table 5: describe in the footnote the statistical test that generated the p-value. Furthermore, I believe that it is more correct here to consider the Mann-Whitney test instead of the Kruskall-Wallis test.
  • Response: Thank you for making this suggestion. We have added the statistical test used to generate the p-values in the footnotes. This can be found on page 14, lines 485-486. With regards to the test used, the statistician in our team suggested the Kruskal-Wallis test and the authors have decided to stay consistent with the Australian study (Hunger for Home Delivery).

Wang, C.; Korai, A.; Jia, S.S.; Allman-Farinelli, M.; Chan, V.; Roy, R.; Raeside, R.; Phongsavan, P.; Redfern, J.; Gibson, A.A.; Partridge, S.R. Hunger for Home Delivery: Cross-Sectional Analysis of the Nutritional Quality of Complete Menus on an Online Food Delivery Platform in Australia. Nutrients 2021, 13, 905. https://doi.org/10.3390/nu13030905

  • Comment 13: Table 5: The authors should clarify how the correction for multiple testing was done (described in the Data analysis section). Are the p-values shown in the table the corrected values
  • Response: The authors have indicated in the data analysis section, what statistical tests were performed for each table in the results section. We have made the appropriate modifications to the data analysis paragraph which can be found on page 6, paragraph 2.4., lines 260-267. Yes, the p-values shown in the table are the corrected values.

Reviewer 2 Report

The article aimed to examine the nutritional quality and marketing attributes of menu items from popular independent and franchise takeaway outlets on New Zealand’s market leading OFD platform (UberEATS® ). The approach is solid and is based on a solid literature review and empirical results resulting from the analysis of a total of 374 popular independent and franchise takeaway outlets. The results showed that the menus from popular local independent and franchise takeaway outlets on Uber Eats  are dominated by menu items that are predominantly unhealthy. These menu items appear to be largely available and exposed to the users through the extensive use of marketing attributes. They are also more likely to appeal to price-sensitive consumers. Special promotions were an additional marketing characteristic that was unique to this study.

The conclusions should be more detailed and linked to the existing literature. 

Author Response

Thank you for allowing us to submit a revised draft of my manuscript titled Unhealthy Food at your Fingertips: Cross-Sectional Analysis of the Nutritional Quality of Takeaway Outlets on an Online Food Delivery Platform in New Zealand to Nutrients. We appreciate the time and effort that you and the reviewers have dedicated to providing your valuable feedback on our manuscript. We are grateful to the reviewers for their insightful comments on my paper. We have been able to incorporate changes to reflect most of the suggestions provided by the reviewers. We have highlighted the changes within the manuscript.

Here is a point-by-point response to the reviewer 2's comments and concerns.

Comments from Reviewer 2

  • Comment 1: The conclusions should be more detailed and linked to the existing literature
  • Response: Thank you for making this suggestion. We have now made modifications to the conclusions paragraph with links to the previous studies, literatures and supporting evidence. This modification can be found on pages 17-18, lines 645-671.

Round 2

Reviewer 1 Report

The authors made the corrections (or argumentations) satisfactorily. Therefore, my decision is to accept the paper.

Author Response

The authors made the corrections (or argumentations) satisfactorily. Therefore, my decision is to accept the paper.

The authors would like to thank the reviewer for reviewing the revised manuscript again and for acknowledging that the authors have made the corrections satisfactorily and the recommendation that the paper can be accepted.